# Resection is responsible for loss of transcription around a double-strand break in *Saccharomyces cerevisiae*

**Nicola Manfrini[1][†], Michela Clerici[1], Maxime Wery[2], Chiara Vittoria Colombo[1], Marc Descrimes[2], Antonin Morillon[2]\*, Fabrizio d'Adda di Fagagna[3,4]\*, Maria Pia Longhese[1]\***

[1]Dipartimento di Biotecnologie e Bioscienze, Università di Milano-Bicocca, Milan, Italy; [2]Institut Curie, Dynamics of Genetic Information: Fundamental Basis and Cancer, Université Pierre et Marie Curie, Paris, France; [3]IFOM Foundation, FIRC Institute of Molecular Oncology Foundation, Milan, Italy; [4]Istituto di Genetica Molecolare, Consiglio Nazionale delle Ricerche, Pavia, Italy

**\*For correspondence:** antonin.
morillon@curie.fr (AM); fabrizio.
dadda@ifom.eu (Fd'AdF);
mariapia.longhese@unimib.
it (MPL)

**Present address:** [†]National
Institute of Molecular Genetics
'Romeo ed Enrica Invernizzi',
Milan, Italy

**Competing interests:** The
authors declare that no
competing interests exist.

**Reviewing editor:** Jessica K
Tyler, University of Texas MD
Anderson Cancer Center, United
States

**Abstract** Emerging evidence indicate that the mammalian checkpoint kinase ATM induces transcriptional silencing *in cis* to DNA double-strand breaks (DSBs) through a poorly understood mechanism. Here we show that in *Saccharomyces cerevisiae* a single DSB causes transcriptional inhibition of proximal genes independently of Tel1/ATM and Mec1/ATR. Since the DSB ends undergo nucleolytic degradation (resection) of their 5′-ending strands, we investigated the contribution of resection in this DSB-induced transcriptional inhibition. We discovered that resection-defective mutants fail to stop transcription around a DSB, and the extent of this failure correlates with the severity of the resection defect. Furthermore, Rad9 and generation of γH2A reduce this DSB-induced transcriptional inhibition by counteracting DSB resection. Therefore, the conversion of the DSB ends from double-stranded to single-stranded DNA, which is necessary to initiate DSB repair by homologous recombination, is responsible for loss of transcription around a DSB in *S. cerevisiae*.

## Introduction

DNA double-strand breaks (DSBs) are particularly dangerous for cells, since their inefficient or inaccurate repair can result in deletions and chromosomal translocations that can lead to cancer and/or severe developmental abnormalities in humans. DSB formation leads to activation of a complex DNA damage response (DDR), whose key players are highly conserved protein kinases, which include human ATM and ATR, as well as their *Saccharomyces cerevisiae* orthologs Tel1 and Mec1 (*Gobbini et al., 2013*). Once activated by DSBs, ATM/Tel1 and ATR/Mec1 promote DSB repair, delay cell cycle progression or trigger the elimination of genetically unstable cells by inducing cell death.

One of the main mechanisms to repair DSBs is homologous recombination (HR), which requires resection of the broken ends in order to generate 3′-ended single-stranded DNA (ssDNA) tails that invade the homologous undamaged template. In *S. cerevisiae*, DSB resection is initiated by the evolutionarily conserved MRX (Mre11-Rad50-Xrs2) complex that, together with Sae2, catalyzes the initial processing of the DSB ends. The 5′-ending strands can then be further degraded by two other machineries depending on Exo1 and Sgs1-Dna2, respectively (*Symington and Gautier, 2011*).

In eukaryotes, the DDR proteins function in the context of a highly organized chromatin environment that needs to be overcome to gain access to damaged DNA. Histone modifications and ATP-dependent chromatin remodelling proteins help to overcome this barrier by altering chromatin structure at the site of damage (*Price and D'Andrea, 2013*). One of the most characterized histone

**eLife digest** DNA is constantly under assault from harmful chemicals; some of which are produced inside the cell, while others come from outside of the cell. Breaks that form across both strands in a DNA double helix are considered the most dangerous type of DNA damage, and can cause a cell to die or become cancerous if they are not repaired accurately.

'Homologous recombination' is one of the main mechanisms used by cells to repair DNA double-strand breaks. This mechanism requires enzymes to eat away at the end of one of the DNA strands on each side of the double-strand break. This process is called 'resection' and it exposes single strands of DNA. These single-stranded DNA 'tails' are then free to interact with an intact copy of the same DNA sequence from elsewhere in the cell's nucleus, which is used as a guide when repairing the damage.

The proteins involved in homologous recombination have to work around other processes that go on inside the nucleus, such as the transcription of DNA in genes into RNA molecules. Previous research has reported that forming a double-strand break in the DNA reduces the levels of transcription for the genes that surround the break, but it was not clear how this occurred.

In mammalian cells, inhibiting the transcription of genes around a double-strand DNA break depends on a signaling pathway that is activated whenever DNA damage is detected. Manfrini et al. now show that this is not the case for budding yeast (*Saccharomyces cerevisiae*). Instead, the experiments indicate that it is the resection of the DNA around a double-strand break to form single-stranded tails that inhibits transcription in budding yeast. One of the next challenges will be to see if the resection process makes any contribution to changes in the transcription of genes that surround a double-strand break in mammals as well.

modifications is the phosphorylation of histone H2AX by ATM/Tel1 and ATR/Mec1, which spreads away from the DSB into large domains of surrounding chromatin (*Rogakou et al., 1999*; *Downs et al., 2000*; *Burma et al., 2001*; *Ward and Chen, 2001*). Other chromatin alterations detected at DSBs are associated with open and decondensed chromatin, indicating that chromatin at DSBs undergoes a transition to a more open, less compact conformation (*Price and D'Andrea, 2013*).

However, several proteins associated with repressive or transcriptionally inactive chromatin, including HP1, PcG (Polycomb group) proteins, PRMD2 methyltransferase, KAP-1, su(var)3-9 methyltrasferase variant (SUV3-9) and the macrohistone variant macroH2A1, are recruited to DNA lesions (*Soria et al., 2012*), suggesting that repressive chromatin can be generated around DSBs. Consistent with this hypothesis, generation of several DSBs distal to the promoter of a reporter gene in mammalian cells leads to ATM-dependent transcriptional repression of this reporter gene (*Shanbhag et al., 2010*), possibly through phosphorylation of the transcriptional elongation factor ENL (*Ui et al., 2015*). Similarly, RNA polymerase I-mediated transcription of rDNA is inhibited in an ATM-dependent manner in the vicinity of DSBs (*Kruhlak et al., 2007*). Furthermore, the steady state RNA levels of genes proximal to a single DSB have been observed to decrease by microarray analysis also in *S. cerevisiae* cells (*Lee et al., 2000*).

However, a different study in mammalian cells, where individual DSBs were induced at discrete endogenous sequences, showed that only transcription of the DSB-containing gene was affected (*Pankotai et al., 2012*). Furthermore, non-coding RNAs that control DDR activation are induced in the surrounding of DSBs in both vertebrates and *Arabidopsis* (*Francia et al., 2012*; *Michalik et al., 2012*; *Wei et al., 2012*), indicating that transcription can occur around DSBs.

Given these apparently contrasting results, other studies are required to understand how DSBs affect transcription in their surroundings. Furthermore, as DSBs are resected to generate 3′-ended ssDNA, and ATM, macroH2A1, PRMD2 and HP1 proteins are required for this process (*Jazayeri et al., 2006*; *Soria and Almouzni, 2013*; *Ayrapetov et al., 2014*; *Khurana et al., 2014*), whether DSB resection has a role in the DSB-induced transcriptional inhibition needs to be investigated.

By using the budding yeast HO endonuclease to create single DSBs at different chromosomal loci, we show that DSB induction causes Mec1- and Tel1-independent transcriptional inhibition of genes surrounding the DSB site. Failure to resect the DSB ends prevents this transcriptional inhibition, which

is instead enhanced by accelerating the resection process. Altogether, these data indicate that loss of transcription around a DSB in *S. cerevisiae* cells is due to the conversion of DSB ends from double-stranded DNA (dsDNA) to ssDNA.

## Results

### Transcription is reduced *in cis* to a DSB at the *MAT* locus

To investigate changes in the transcription of genes surrounding DSBs, we took advantage of a yeast haploid strain (JKM139) where a single DSB can be generated at the *MAT* locus by galactose-induced expression of the HO endonuclease. As the homologous donor sequences *HML* and *HMR* are deleted in this strain, this DSB cannot be repaired by HR (*Lee et al., 1998*). We previously used this strain for total RNA-seq analysis of protein-coding gene expression upon induction of the HO-induced DSB to show that mRNA levels of the vast majority of protein-coding genes underwent no significant change upon DSB generation (*Manfrini et al., 2015*). Here we focused on the *MAT* locus that was not analyzed in the previous study. We mapped the RNA-seq data from two biological replicates of the JKM139 wild-type strain grown in raffinose (time zero, T0) and shifted to galactose-containing medium for 60 (T60) and 240 (T240) min on a restricted reference 'genome' corresponding to the *MAT* locus ±10 kb (*Figure 1A*). Quantitative analysis of tag densities for annotated genes in this region revealed a moderate signal decrease for the DSB-proximal genes 60 min after HO induction, and a strong signal reduction along the whole region after 240 min (*Figure 1B*). Strikingly, after 240 min of HO induction, density for the proximal genes *BUD5*, *HMRA2*, *HMRA1* and *TAF2* showed a ≥10-fold decrease compared to time zero, while density was only reduced by about twofold for the distal genes *SNT1*, *IMG1* and *BUD23* (*Figure 1C*).

Validation of the above data by using quantitative reverse transcriptase PCR (qRT-PCR) confirmed the reduction in transcript levels for genes located around the HO-induced DSB 240 min after HO induction (*Figure 2A*). The level of these mRNAs decreased progressively as the distance of the corresponding genes from the DSB diminished and such decrease was independent of the strand that was transcribed (*Figure 2A*). These decreases measured 240 min after HO induction were not influenced by mRNA stability, as all the analyzed mRNAs have half-lives shorter than 50 min (*Geisberg et al., 2014*).

Because both qRT-PCR and RNA-seq measured the steady state level of total RNAs, which is a balance between synthesis and degradation, we investigated whether the reduction of RNA levels for genes around the DSB was due to transcriptional inhibition. To this end, the binding of the Rpb2 second largest subunit of RNA polymerase II was measured at T0 and 240 min after HO induction at different distances from the DSB. Indeed, Rpb2 association in the surroundings of the DSB decreased and this effect progressively diminished as a function of the distance from the DSB (*Figure 2B*), indicating that transcription was inhibited specifically around the DSB.

### Local transcriptional inhibition is a general response to DSB formation

To investigate whether the DSB-induced transcriptional inhibition was specific for the *MAT* locus, we analyzed RNA levels and RNA polymerase II binding in tGI354 and YFP17 strains, which carried the recognition site for the HO endonuclease at the *ARG5,6* locus on chromosome V or at the *LEU2* locus on chromosome III, respectively. Since tGI354 strain can use the uncleavable *MATa-inc* sequence on chromosome III as a donor to repair the HO-induced DSB by Rad51-dependent HR, this strain carried the deletion of *RAD51*. After generation of the HO-induced DSB at the *ARG5,6* locus, we found that both the RNA levels of genes located in the surrounding of the DSB (*Figure 3A*) and Rbp2 occupancy (*Figure 3B*) progressively decreased as the distance of the corresponding genes from the DSB diminished. Similar results were obtained when the HO-induced DSB was generated at the *LEU2* locus (*Figure 3C,D*). We conclude that inhibition of local transcription is a general response to DSB formation.

### The checkpoint kinases Tel1 and Mec1 are not required for DSB-induced transcriptional inhibition

In all eukaryotes, DNA DSBs induce a DDR that depends on the checkpoint kinases ATM/Tel1 and ATR/Mec1 (*Gobbini et al., 2013*). As transcriptional repression in proximity to DSBs

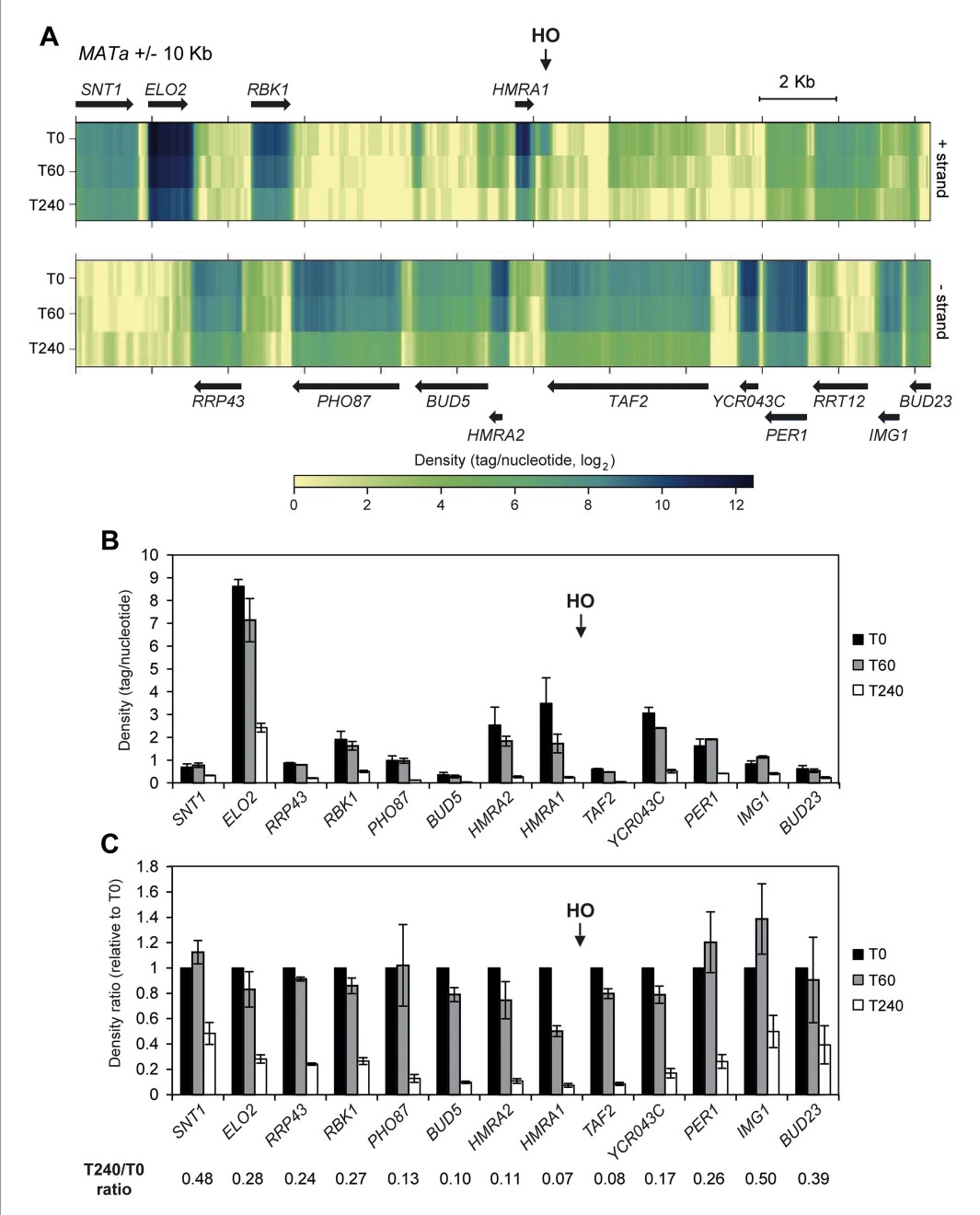

**Figure 1**. Decrease of RNA levels around a DNA double-strand break (DSB). (**A**) Strand-specific RNA-seq data from the two biological replicates of wild type strain (JKM139) before (time zero, T0) and 60 (T60) or 240 (T240) min after HO induction, were uniquely mapped to the *MAT* locus ±10 kb. For each time point, densities (tag/nucleotide, log₂ scale) from the two replicates were pooled and visualized as a heatmap with the upper and lower panels corresponding to the + and − strands, respectively. Black arrows represent annotated genes. (**B**) Density for genes along the *MAT* locus ±10 kb at T0, T60 and T240 after HO induction. Mean values ±s.d. were calculated from the two biological replicates analyzed. *RRT12* showed too low signal to be significantly quantified and was not included. (**C**) Ratio of density for genes along the *MAT* locus ±10 kb at T0, T60 and T240 after HO induction. For each time point, densities were normalized on the values obtained at T0. Mean values ±s.d. were calculated as above. RNA-seq data used to construct the graphs of *Figure 1B,C* are available in *Figure 1—source data 1*.

The following source data is available for figure 1:

**Source data 1**. RNA-seq data used to construct the graphs of *Figure 1B,C*.

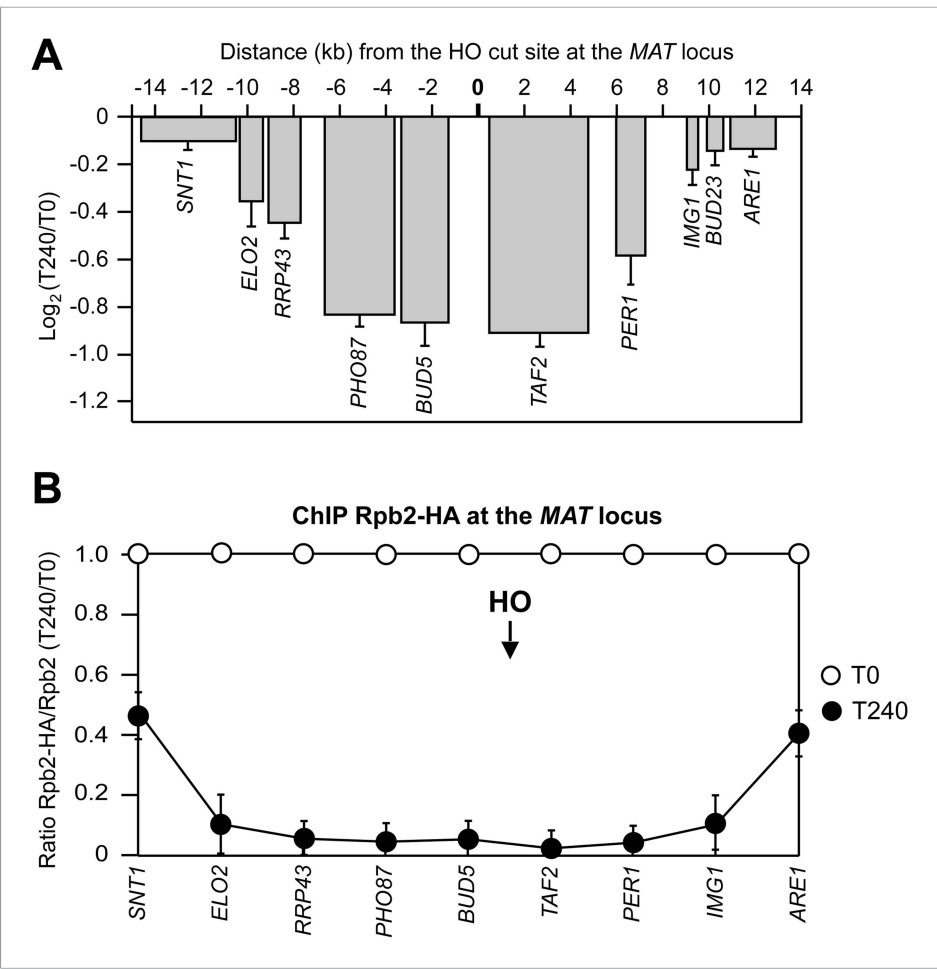

**Figure 2**. DSB-induced transcriptional inhibition at the *MAT* locus. (**A**) YEPR exponentially growing cell cultures of the JKM139 strain, carrying the HO cut site at the *MAT* locus, were transferred to YEPRG at T0 to induce HO. RNA levels of genes located at different distances from the HO cut site were evaluated by quantitative reverse transcriptase PCR (qRT-PCR) at T0 and T240 after HO induction. Results are presented as ratios between T240 and T0. RNA levels were quantified using ΔΔCt method and quantities were normalized to *ACT1* RNA levels. The mean values ±s.d. are represented (n = 3). (**B**) Exponentially growing YEPR cell cultures of the JKM139 derivative strains expressing either a fully functional Rpb2-HA fusion protein or untagged Rpb2 were transferred to YEPRG at T0 to induce HO. Binding of Rpb2-HA at different distance from the DSB at T0 and T240 after HO induction was evaluated by ChIP and qPCR. Primers used were the same as in (**A**). Results are presented as ratios between Rpb2-HA and untagged Rpb2, both of which normalized against the corresponding input, at T240 relative to T0. The mean values ±s.d. are represented (n = 3). Rpb2-HA binding at the *ACT1* gene was used as internal control.

requires ATM in mammalian cells (***Kruhlak et al., 2007***; http://journal.frontiersin.org/Journal/ 10.3389/fgene.2013.00136/full, ***Shanbhag et al., 2010***), we asked whether Tel1 and/or Mec1 were necessary for transcriptional silencing of genes flanking the HO-induced DSB. HO expression was induced by galactose addition in *mec1Δ*, *tel1Δ* and *mec1Δ tel1Δ* cells (*mec1Δ* and *mec1Δ tel1Δ* cells were kept viable by *SML1* deletion) and mRNA levels were analyzed 240 min after HO induction. As the lack of Mec1 causes cells to progress through the cell cycle even after DSB formation, HO was induced in cells arrested in G2 with nocodazole and kept arrested in G2 during HO-cut induction. As shown in ***Figure 4***, the amount of RNA transcribed from genes surrounding the DSB was reduced to quite similar extents in HO-induced wild type, *mec1Δ sml1Δ*, *tel1Δ* and *mec1Δ tel1Δ sml1Δ* cells, with *mec1Δ sml1Δ* cells showing a more pronounced decrease. This finding indicates that neither Tel1 nor Mec1 are responsible for DSB-induced transcriptional inhibition.

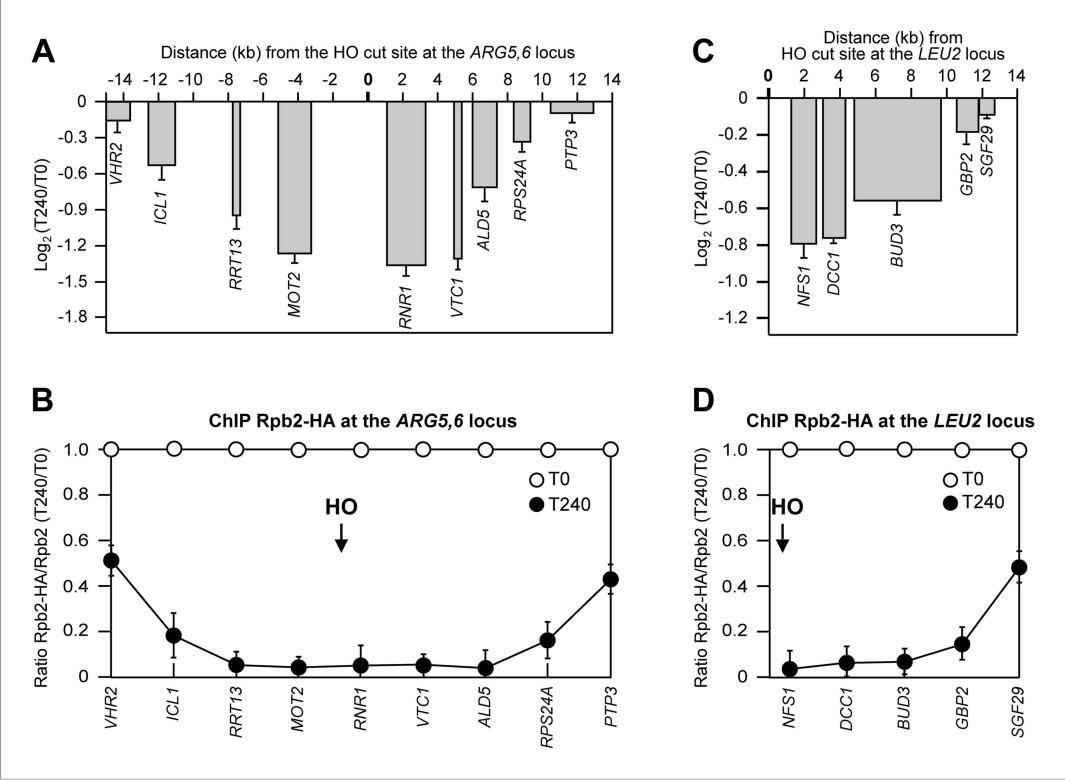

**Figure 3**. DSB-induced transcriptional inhibition at different chromosomal loci. (**A**) YEPR exponentially growing cell cultures of tGI354 strain, carrying the HO site at the *ARG5,6* locus and the deletion of *RAD51* gene, were transferred to YEPRG at T0 to induce HO. RNA levels of genes located at different distances from the HO cut site were evaluated by qRT-PCR at T0 and T240 after HO induction, as described in *Figure 2A*. Results are presented as ratios between T240 and T0. The mean values ±s.d. are represented (n = 3). (**B**) Binding of Rpb2-HA from samples collected in (**A**) was evaluated by ChIP and qPCR as described in *Figure 2B*. Primers used were the same as in (**A**). The mean values ±s.d. are represented (n = 3). (**C**) YEPR exponentially growing cell cultures of YFP17 strain, carrying the HO cut site at the *LEU2* locus, were transferred to YEPRG at T0 to induce HO. RNA levels were analyzed as in *Figure 2A* but normalized to the *ALG9* gene transcript. Only transcription of genes located on the right side of the HO-induced DSB was analyzed because no transcription units are present on the left side of the break. The mean values ±s.d. are represented (n = 3). (**D**) Binding of Rpb2-HA from samples collected in (**C**) was evaluated by ChIP and qPCR as described in *Figure 2B*. Primers used were the same as in (**C**).

## γH2A and Rad9 limit DSB-induced transcriptional inhibition by negatively regulating DSB resection

A single DSB triggers phosphorylation of histone H2A to form γH2A, which spreads on each side of the DSB to about 50 kb distance (*Rogakou et al., 1999*; *Shroff et al., 2004*). Interestingly, γH2A diminishes strongly over highly transcribed regions and H2A phosphorylation is rapidly restored if transcription is subsequently inhibited (*Lee et al., 2014*). We therefore investigated whether γH2A plays a role in DSB-induced transcriptional inhibition by expressing HO in *hta1-S129A* cells, which produced a H2A variant carrying a non-phosphorylatable alanine residue replacing Ser129. As histone H2A is encoded by the two genes *HTA1* and *HTA2*, *hta1-S129A* cells also carried the *HTA2* deletion. Strikingly, not only RNA levels of genes proximal to the DSB still decreased in *hta1-S129A* cells, but this decrease was more severe than in wild type cells (*Figure 5A*), indicating that γH2A counteracts repression of transcription around the DSB.

The ends of DSBs are nucleolytically processed to generate 3′-ended ssDNA that initiate HR. DSB resection is negatively regulated by the checkpoint protein Rad9, which inhibits mainly the Sgs1-Dna2 resection machinery (*Bonetti et al., 2015*; *Ferrari et al., 2015*). The lack of γH2A reduces the amount of Rad9 bound at the DSB ends and, as a consequence, *hta1-S129A* cells resect the DSB ends more

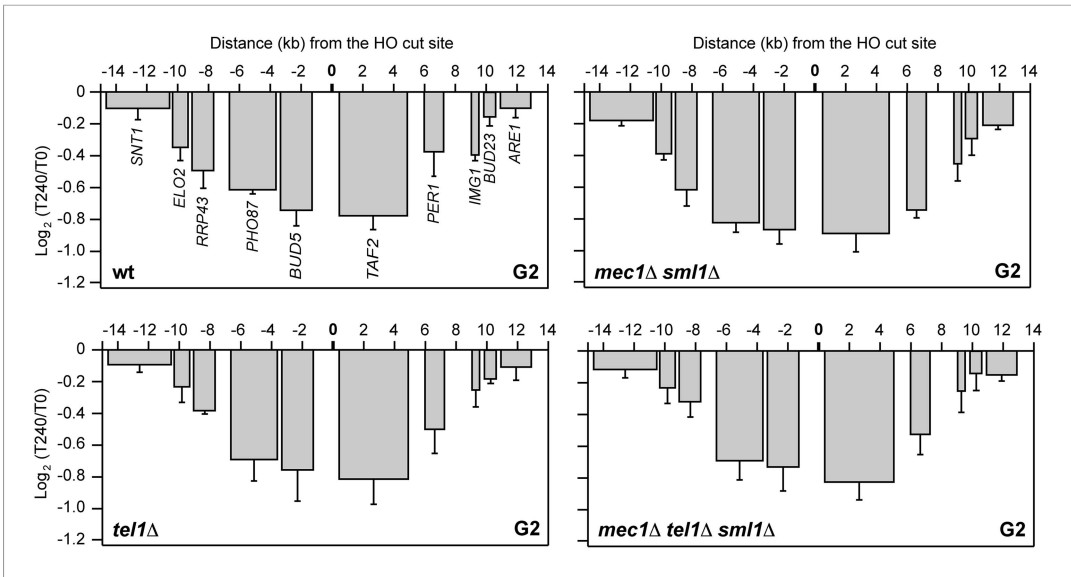

**Figure 4**. DSB-induced transcriptional inhibition does not require Mec1 and Tel1. YEPR exponentially growing cell cultures of the JKM139 strain, carrying the HO cut site at the *MAT* locus, were arrested in G2 with nocodazole and transferred at T0 to YEPRG in the presence of nocodazole. RNA levels were analyzed by qRT-PCR as described in *Figure 2A*. The mean values ±s.d. are represented (n = 3).

efficiently than wild type cells (*Eapen et al., 2012*; *Clerici et al., 2014*). Thus, we investigated whether the enhanced DSB-induced transcriptional repression observed in *hta1-S129A* mutant cells was due to their accelerated nucleolytic processing of the DSB ends. Resection of the DSB 5′ strand can be measured by following the loss of restriction fragments by Southern blot analysis with a ssRNA probe annealing on one side of the break (*Figure 5B*). Strikingly, *SGS1* deletion, which reduced the accelerated resection in *hta1-S129A* cells to almost wild type levels (*Figure 5C*), also reduced their enhanced DSB-induced transcriptional inhibition (*Figure 5A*). Furthermore, *rad9Δ* cells, which undergo accelerated DSB resection (*Lazzaro et al., 2008*), showed transcription inhibition around the DSB to the same extent as *hta1-S129A* cells (*Figure 5A*). Altogether these data indicate that γH2A and Rad9 limit the inhibition of transcription around the DSB ends by counteracting the resection process.

## Loss of transcription around a DSB depends on resection

We then investigated if generation of ssDNA was responsible for transcription inhibition at DSBs. Resection has been reported to initiate asynchronously from the ends of DSBs (*Shroff et al., 2004*) and to move about 4 kb/hr (*Vaze et al., 2002*). After 60 min of HO induction, wild type cells accumulated mostly r2 resection products, indicating that DSB resection in most cells had not proceeded beyond 3.5 kb from the DSB (*Figure 6A*). At the same time point, we detected a strong decrease of Rpb2 occupancy at the *TAF2* and *BUD5* genes, which are located within 4 kb from either side of the DSB (*Figure 6B*). Concomitantly with the progression of 5′–3′ resection (*Figure 6A*), also the binding of Rpb2 to distal genes progressively diminished (*Figure 6B*), suggesting that the decrease of RNA polymerase II occupancy around the DSB correlates with the time it takes for a DNA-end to become single-stranded.

DSB resection in *S. cerevisiae* is initiated by MRX and Sae2, which catalyze an endonucleolytic cleavage of the 5′ strands. The resulting partially resected 5′ strand can be further processed by the nucleases Exo1 and Dna2, the latter working in concert with the 3′–5′ helicase Sgs1 (*Mimitou and Symington, 2008*; *Zhu et al., 2008*). DSB resection is severely affected by either simultaneous inactivation of Exo1 and Sgs1 or the lack of MRX, with the latter not only providing the endonuclease activity to initiate resection, but also promoting the loading of Exo1, Sgs1 and Dna2 at DSB ends (*Shim et al., 2010*). To assess whether DSB resection has a role in the DSB-induced transcriptional

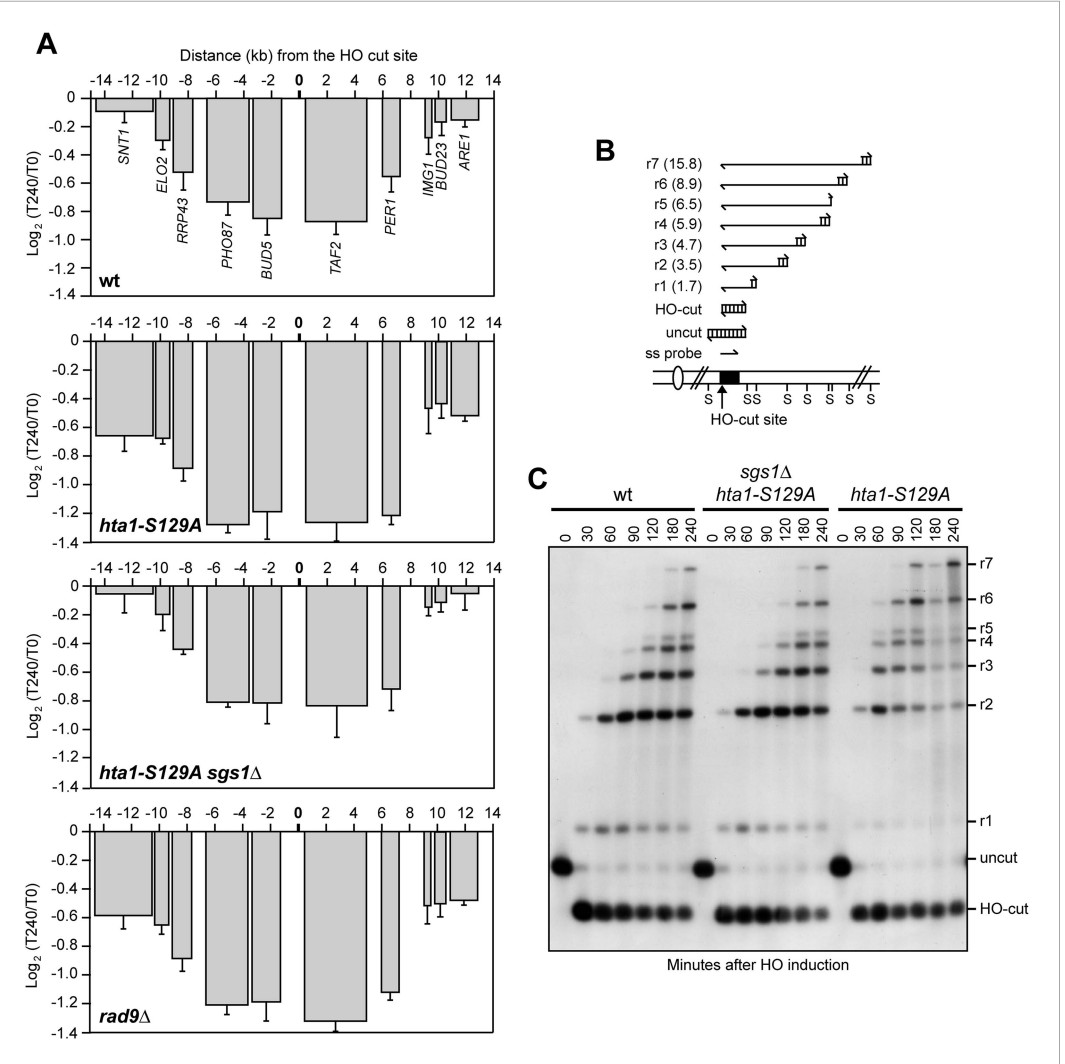

**Figure 5.** The lack of γH2A enhances transcriptional inhibition around the DSB by accelerating resection. (**A**) YEPR exponentially growing cell cultures of the JKM139 derivative strains, carrying the HO cut site at the *MAT* locus, were transferred to YEPRG at T0. RNA levels of genes located in the surroundings of the HO cut site at the *MAT* locus were analyzed at T0 and T240 after HO induction by qRT-PCR as described in *Figure 2A*. The mean values ±s.d. are represented (n = 3). (**B**) Method to measure DSB resection. Gel blots of SspI-digested genomic DNA separated on alkaline agarose gel were hybridized with a single-stranded *MAT* probe (ss probe) that anneals to the unresected strand. 5′–3′ resection progressively eliminates SspI sites (S), producing larger SspI fragments (r1 through r7) detected by the probe. (**C**) DSB resection. Genomic DNA prepared from samples collected in (**A**) was analysed for single-stranded DNA (ssDNA) formation at the indicated times after HO induction as described in (**B**). The image that was used for the cropped final *Figure 5C* is available in *Figure 5—source data 1*.

The following source data is available for figure 5:

**Source data 1**. Image that was used for the cropped final *Figure 5C*.

silencing, we asked whether *exo1Δ*, *mre11Δ* or *exo1Δ sgs1Δ* cells, which were impaired in resection to different extents, were still capable to repress transcription of genes in the vicinity of the HO-induced DSB. Indeed, neither the amount of Rpb2 bound around the HO-induced DSB (*Figure 6C*) nor RNA levels were decreased in *exo1Δ sgs1Δ* cells (*Figure 6D*), which had the most severe DSB resection defect compared to *mre11Δ* and *exo1Δ* single mutants (*Figure 6A*). RNA levels only slightly decreased in HO-induced *mre11Δ* cells (*Figure 6D*), which were severely

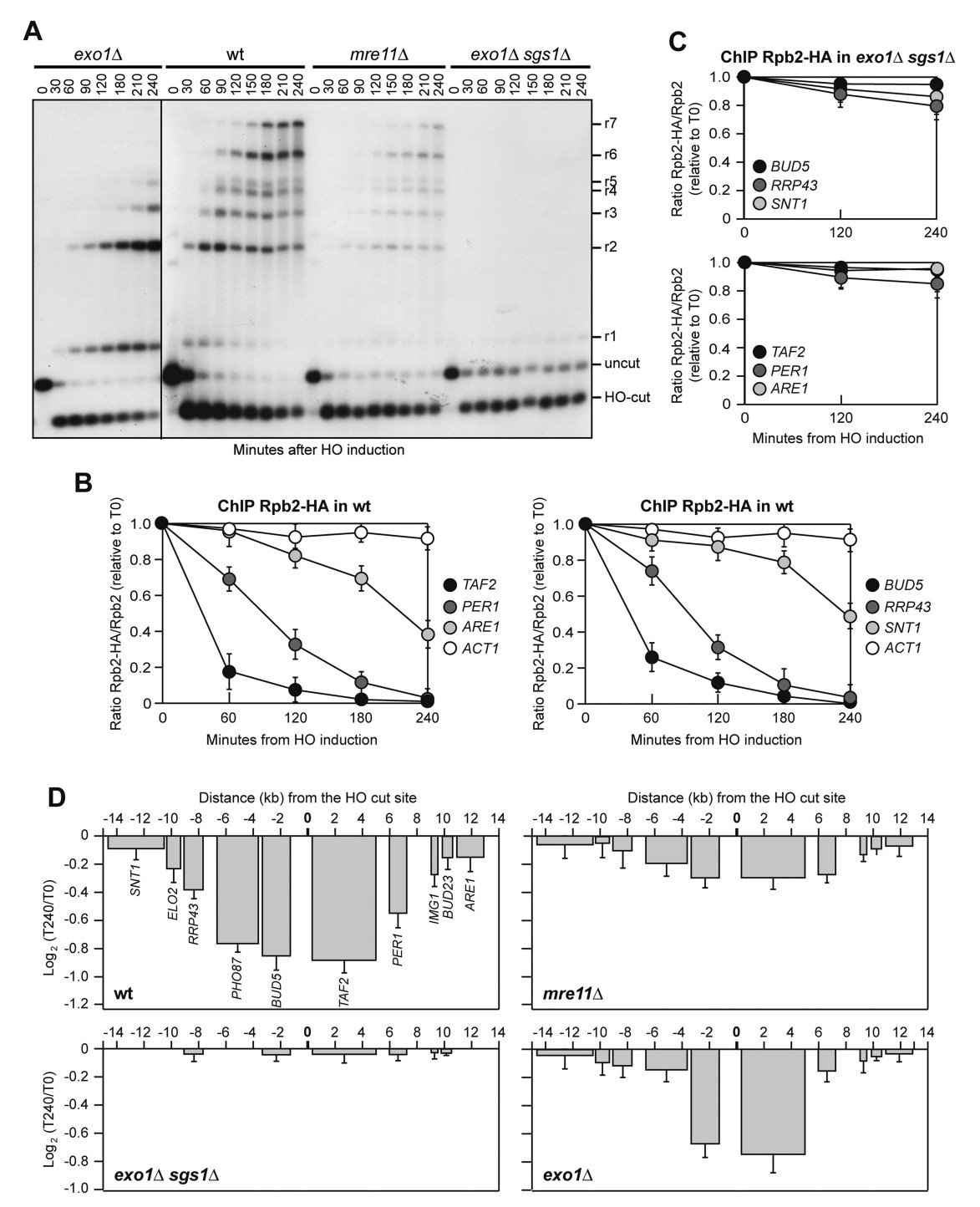

**Figure 6**. Resection mutants fail to reduce transcription around the DSB. (**A**) DSB resection. YEPR exponentially growing cell cultures of JKM139 derivative strains, carrying the HO cut site at the *MAT* locus, were transferred to YEPRG at T0. Formation of ssDNA was determined as described in *Figure 5B*. (**B**) Binding of Rpb2-HA in wild type cells from samples collected in (**A**) was evaluated by ChIP and qPCR as described in *Figure 2B*. The mean values ±s.d. are represented (n = 3). (**C**) Binding of Rpb2-HA in *exo1Δ sgs1Δ* cells from samples collected in (**A**) was evaluated by ChIP and qPCR as described in *Figure 2B*. The mean values ±s.d. are represented (n = 3). (**D**) RNA levels from samples collected in (**A**) were analyzed at T0 and T240 after HO induction by qRT-PCR as described in *Figure 2A*. The mean values ±s.d. are represented (n = 3). Contiguous images that were used for the cropped final *Figure 6A* are available in *Figure 6—source data 1*.

The following source data is available for figure 6:

**Source data 1**. Contiguous images that were used for the cropped final *Figure 6A*.

defective in the accumulation of resection products (*Figure 6A*), while only the RNA levels of genes proximal to the break site were diminished in *exo1Δ* mutant cells (*Figure 6D*), which initiated DSB resection but failed to accumulate resection products longer than 3.5 kb (*Figure 6A*). Thus, loss of transcription around a DSB in *S. cerevisiae* is due to the conversion of dsDNA to ssDNA by the resection machinery.

## Discussion

Here, we further extend the observation that induction of a DSB at the *MAT* locus in *S. cerevisiae* cells leads to a decrease of the RNA steady state levels of proximal genes (*Lee et al., 2000*), by demonstrating that this decrease is not locus specific and is due to dissociation of RNA polymerase II from its template DNA. In contrast to the reported requirement of mammalian ATM for transcription silencing around a DSB (*Kruhlak et al., 2007*; *Shanbhag et al., 2010*; *Ui et al., 2015*), neither Tel1 nor Mec1 appear to be necessary for transcriptional inhibition at the DSB site in *S. cerevisiae*. Instead, we found that mutants defective in DSB resection fail to inhibit transcription around a DSB, and the extent of this failure correlates with the severity of the DSB resection defect. Furthermore, accelerating DSB resection by preventing γH2A generation or by eliminating Rad9 enhances this DSB-induced transcriptional inhibition. Altogether, these data indicate that loss of transcription around a DSB in *S. cerevisiae* is not due to an active regulatory mechanism, but to the conversion of the DSB ends from dsDNA to ssDNA.

Nucleolytic processing of the template DNA strand should stop transcription of the corresponding gene. However, we found that mRNA levels and RNA polymerase II occupancy around a DSB decrease independently of the strand that is transcribed, indicating that loss of non-template DNA strands also impedes transcription. Therefore, the most likely reason of why the conversion from dsDNA to ssDNA stops transcription is that the transcription machinery binds only dsDNA. Consistent with this hypothesis, it has been shown that within a region containing a stretch of dsDNA preceding a single-strand 3′ DNA end, purified RNA polymerase II transcribes within the dsDNA portion (*Lilley and Houghton, 1979*; *Kadesch and Chamberlin, 1982*). Furthermore, structural analyses of transcription initiation reveals that RNA polymerase II and its associated General Transcription Factors bind double-stranded promoter DNA to form a closed preinitiation complex (PIC), where the transcriptional machinery interacts with both template and non-template DNA strands (*Liu et al., 2013*; *Sainsbury et al., 2015*). Moreover, removal of either the template or non-template strands prevents binding of T7 RNA polymerase to the promoter, supporting the importance of protein-dsDNA interaction in the spatial organization of the transcriptional initiation complex (*Maslak and Martin, 1993*).

Interaction of the transcriptional machinery with dsDNA also allows transcription initiation and elongation. In fact, the double-stranded promoter DNA follows a straight path in the PIC complex and this rigidity allows the subsequent transition from a closed to an open promoter complex, where a central DNA region is melted leading to a transcription bubble in which the DNA template strand enters the RNA polymerase II cleft (*Murakami et al., 2013*). Furthermore, competition between the non-template DNA strand and the RNA transcript for base-pairing with the DNA template strand was shown to be important for maintaining structure and function during elongation (*Kireeva et al., 2011*), reinforcing the inhibitory effects of ssDNA on the transcription process.

In summary, while DSB-induced transcriptional repression in mammals is reportedly an active mechanism controlled by ATM, the stop of transcription around a DSB in *S. cerevisiae* cells is due to the conversion of dsDNA to ssDNA that is necessary to initiate DSB repair by HR (*Figure 7*). In any case, while Mec1 and Tel1 are not required to resect a DSB, ATM and the chromatin compaction promoting proteins macroH2A1, PRMD2 and HP1 promote end resection by facilitating the loading of BRCA1 at the DSB ends (*Soria and Almouzni, 2013*; *Ayrapetov et al., 2014*; *Khurana et al., 2014*). Thus, whether the nucleolytic processing of the DSB ends contributes to repress transcription around a DSB also in mammals is an important question that remains to be addressed. In fact, although in human cells inactivation of the end resection factor CtIP does not restore transcription around the DSB (*Shanbhag et al., 2010*), other resection machineries are known to be active in resecting DSBs in CtIP-depleted cells (*Tomimatsu et al., 2012*) and could contribute to inhibit transcription around the DSB.

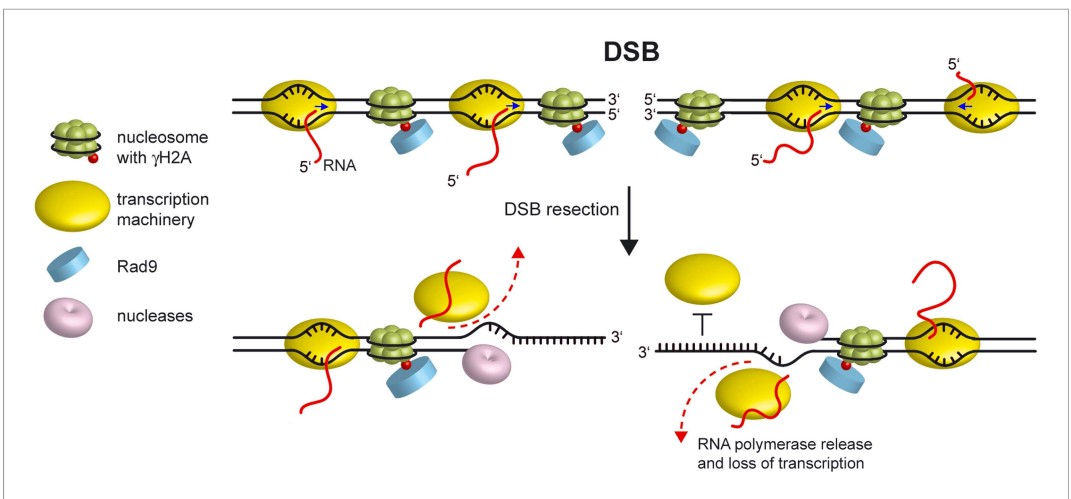

**Figure 7**. Model for loss of transcription around a DSB. Resection of the 5′ strands at both DSB ends leads to release of the transcription machinery (dashed lines) and to subsequent transcription arrest independently of whether the degraded DNA strand acts as template or non-template. Since the RNA polymerase binds double-stranded promoter DNA, generation of ssDNA at the DSB ends prevents reinitiation events (bar-headed line). Blue arrows indicate direction of transcription.

## Materials and methods

### Yeast strains
The yeast strains used in this study are derivatives of JKM139, YFP17 or tGI354 strains and are listed in *Supplementary file 1*. Cells were grown in YEP medium (1% yeast extract, 2% peptone) supplemented with 2% glucose (YEPD), 2% raffinose (YEPR) or 2% raffinose and 3% galactose (YEPRG).

### DSB resection
DSB end resection at the *MAT* locus in JKM139 derivative strains was analyzed on alkaline agarose gels as previously described (*Clerici et al., 2008*), by using a single-stranded probe complementary to the unresected DSB strand. This probe was obtained by in vitro transcription using Promega (Madison, WI) Riboprobe System-T7 and plasmid pML514 as a template. Plasmid pML514 was constructed by inserting in the pGEM7Zf EcoRI site a 900-bp fragment containing part of the *MATα* locus (coordinates 200870 to 201587 on chromosome III).

### Total RNA-seq analysis
Total RNA-seq libraries were previously described (*Manfrini et al., 2015*) and data were retrieved from the Gene Expression Omnibus (accession number GSE63444; *Manfrini et al., 2014*). Reads were uniquely mapped as described (*Manfrini et al., 2015*) on a reference 'genome' corresponding to the *MATa* sequence ±10 kb, that was manually reconstructed by replacing *MATα*-specific elements with the corresponding *MATa*-specific elements in the *MATα* locus sequence ±10 kb (chr. 3, positions 188671 to 211177) retrieved from SGD (http://www.yeastgenome.org/). Data were normalized using the normalization factors previously used for the whole transcriptome analysis (*Manfrini et al., 2015*), based on the total number of reads that mapped on all the ORFs of the yeast genome.

### qRT-PCR
Total RNA was extracted from cells using the Bio-Rad (Hercules, CA) Aurum total RNA mini kit. First strand cDNA synthesis was performed with the Bio-Rad iScript cDNA Synthesis Kit. qRT-PCR was performed on a MiniOpticon Real-time PCR system (Bio-Rad) and RNA levels were quantified using the ΔΔCt method. Quantities were normalized to either *ACT1* or *ALG9* RNA levels. Since *RNR1* is

transcriptionally induced immediately after HO-induced DSB, its RNA level at T240 was normalized on the value obtained 30 min after HO induction, when DSB resection was not proceeded beyond 1.7 kb. Primer sequences are provided in *Supplementary file 2*.

### ChIP analysis

ChIP analysis was performed as previously described (*Viscardi et al., 2007*). Chromatin extracts from both *RPB2-HA* and *RPB2* strains were immunoprecipitated with anti-HA antibodies (12CA5). Input and immunoprecipitated DNA were purified and analyzed by qPCR. Amplicons were chosen well within the coding region of the genes and of the highly transcribed *ACT1* gene on chromosome VI as a control. The ratio between values obtained for Rpb2-HA and those obtained with the untagged Rpb2 immunoprecipitated samples, both of which normalized against the corresponding input, was calculated for each time point after HO induction. The obtained values were divided by the ratio calculated from uninduced cells, arbitrarily set to 1, for each amplicon. Primer sequences are provided in *Supplementary file 2*.

## Acknowledgements

We thank J Haber and G Liberi for strains and G Lucchini for critical reading of the manuscript. High-throughput sequencing was performed by the NGS platform of Institut Curie, supported by the grants ANR-10-EQPX-03 and ANR10-INBS-09-08 from the Agence Nationale de la Recherche (investissements d'avenir) and by the Canceropôle Ile-de-France. NM was supported by a fellowship from Fondazione Italiana per la Ricerca sul Cancro (FIRC).

## Additional information

### Funding

| Funder | Grant reference | Author |
|---|---|---|
| Associazione Italiana per la Ricerca sul Cancro | IG15210 | Maria Pia Longhese |
| PRIN 2010-2011 | | Maria Pia Longhese |
| Agence Nationale de la Recherche | REGULncRNA | Antonin Morillon |
| European Research Council (ERC) | EpincRNA Starting grant | Antonin Morillon |
| European Research Council (ERC) | DARK Consolidator grant | Antonin Morillon |
| Associazione Italiana per la Ricerca sul Cancro | IG12971 | Fabrizio d'Adda di Fagagna |
| Human Frontier Science Program (HFSP) | RGP0014/2012 | Fabrizio d'Adda di Fagagna |
| Fondazione Cariplo | 2010.0818 | Fabrizio d'Adda di Fagagna |
| Fondazione Telethon | GGP12059 | Fabrizio d'Adda di Fagagna |
| PRIN 2010-2011 | | Fabrizio d'Adda di Fagagna |
| Istituto Nazionale Genetica Molecolare | MIUR EPIGEN | Fabrizio d'Adda di Fagagna |
| European Research Council (ERC) | 322728 | Fabrizio d'Adda di Fagagna |

The funders had no role in study design, data collection and interpretation, or the decision to submit the work for publication.

### Author contributions

NM, MC, MW, Conception and design, Acquisition of data, Analysis and interpretation of data; CVC, MD, Acquisition of data, Analysis and interpretation of data; AM, Fd'AdF, MPL, Conception and design, Analysis and interpretation of data, Drafting or revising the article

## Additional files

### Supplementary files

• Supplementary file 1. *Saccharomyces cerevisiae* strains used in this study.

• Supplementary file 2. Primer sequences used in this study.

### Major dataset

The following previously published dataset was used:

| Author(s) | Year | Dataset title | Dataset ID and/or URL | Database, license, and accessibility information |
|---|---|---|---|---|
| Manfrini N, Trovesi C, Wery M, Martina M, Cesena D, Descrimes M, Morillon A, d'Adda di Fagagna F, Longhese MP | 2014 | RNA processing proteins regulate Mec1/ATR activation by promoting generation of RPA-coated ssDNA | http://www.ncbi.nlm.nih.gov/geo/query/acc.cgi?acc=GSE63444 | Publicly available at the NCBI Gene Expression Omnibus (Accession no: GSE63444). |

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
