## [Decision Letter]

Thank you for submitting your work entitled “Resection drives transcriptional repression around a DNA double-strand break in *Saccharomyces cerevisiae*” for peer review at *eLife.* Your submission has been favorably evaluated by Jim Kadonaga (Senior Editor) and three reviewers, one of whom is a member of our Board of Reviewing Editors.

The reviewers have discussed the reviews with one another, and the Reviewing Editor has drafted this decision to help you prepare a revised submission.

Summary:

We all felt that this was an important and high quality study that clearly supported the novel finding that transcription stops due to DNA resection.

1) We all felt that the likely reason why transcription stops and RNA polymerase II occupancy is lost upon resection is because the transcription machinery, such as TBP and RNA polymerase II bind to dsDNA and not ssDNA. This stopping transcription due to loss of DNA template model is quite a different model from the current repression model, which implies an active regulatory mechanism. As such, an in depth discussion of loss of transcription as a consequence of loss of DNA template with appropriate citations, and a model figure to illustrate this idea, is needed.

2) The data generated from the HO site at *LEU2* and the HO site at *MAT* need to be in two separate figures, and the difference emphasized more strongly in the text. Otherwise this will all be assumed to be at *MAT* and the apparent novelty of this work, compared to the previous Lee et al. paper, will be reduced.

3) The result with the *mec1∆ tel1∆* double mutant needs to be reported in order to conclude that the DNA damage checkpoint is not required for stopping transcription around a DNA double strand break.

Reviewer #1:

Although brief, this is an important study that makes significant advances on the previous work from Jim Haber's lab showing that there is progressive transcriptional repression around the DNA double strand break (DSB) induced by the HO endonuclease at the yeast *MAT* locus. There are two important novel results from the current work: 1) the transcriptional repression around a DSB is not dependent on checkpoint activation (in contrast to the situation in mammalian cells); 2) transcriptional repression is dependent on DNA resection. This last finding is very novel and exciting, and has not yet been examined in other systems. The work is of high quality, but the following points need to be addressed:

1) Given the partial redundancy of Tel1 and Mec1 in activation of the DNA damage checkpoint, it is important that they show that transcriptional repression around the HO site is not affected in a *tel1 mec1 sml1* mutant, in order to make the conclusion that the activation of the DNA damage checkpoint is not required for transcriptional repression.

2) Mechanically, it makes sense that DNA resection would block transcription, given that RNA polymerase II binds to double strand DNA, and cannot transcribe from single strand DNA. This should be discussed briefly.

*Reviewer #2*:

This is a very straightforward analysis of a consequence of DNA double-strand break on local transcription. There are a plethora of observations in the literature that bare on this issue, motivating the authors to carry out these studies. At the risk of oversimplification, the message from this story seems to be very simply simplified, that conversion of DNA from double strand to single strand, that allows homologous recombination, has an ‘unintended’ consequence; genes that become single-stranded can no longer by transcribed (presumably it’s well know that RNA polymerase binding and regulation etc. occurs only on double-strand templates, and not on single-strand templates). All of their lovely experiments are consistent with this simple idea, an idea they only state once and, in my view, deserves to be highlighted in the Abstract, and supported with a model at manuscript’s end. Sometimes biology does have simple explanations; this seems to be one of those times.

I have but a few issues I think important to address, most in the writing. The experiments appear impeccable, thorough, clearly illustrated.

1) They need to emphasize that they carried out this analysis on a non-*MAT* locus DSB, Figure 2 is of an HO break at the *LEU2* locus. So, they have tested 2 DSBs. Is their general conclusion made any more convincing by test of a third DSB? Yet, showing data at the HO site at *LEU2* is critical for the generality of their conclusions. I suggest they therefore put the *LEU2* HO data in a separate Figure. Shape the data in a similar fashion as the HO data, state distance from HO site, extending left and right from the DSB, as they did for the *MAT* HO site data. Parallel presentations make it easier for the reader.

2) As I said in the general comments, they need to show a model. The ‘take home’ message is simple, and easily grasped by all with a model. That model can have Rad9 and H2A∆ in it. Yet, most critically is the double-strand to single-strand conversion as the mechanistic explanation. And this inhibition appears to be perhaps unintended consequences of a separate biological reaction, generating a substrate for recombination.

3) They need a citation that in fact single-strand templates do not support transcription.

4) I do not quite get how the transcriptional repression cannot be Mec1 regulated, if Mec1 regulates degradation by regulating Rad9, etc.? And degradation of telomere proximal DNA has complex interactions with Mec1 and Rad53, and Rad9; a word on how the telomere resection is or is not relevant here.

5) In the Results section, the authors allude to a previous paper ([24], EMBO Rep) stating: “We previously used this strain for total RNA-seq analysis of protein-coding gene expression upon induction of the HO-induced DSB”. What did they conclude from that study relevant to this study?

6) The conclusion that the inhibition is not cell cycle regulated, based on but two experiments – asynchronous and G2/M synchronized – is underwhelming. At least under those two conditions they see no difference; tone down the generalization, as the experiments do not support that conclusion. I do not suggest getting rigorous about the cell cycle effects, or lack thereof; this is not an important issue, it seems. Just have the language match the data in this point.

7) Make it more transparent how gene transcription reduction correlates with extent of degradation. So, when reporting on the r2 restriction site, state how far away it is from the HO site, what gene is there, and how much that genes transcription is reduced and at what time.

8) At the Results outset, I think it best to set the reader up for the story and technical issues. It is relevant that the time scale, in hours, is far longer than the half-lives of the mRNAs involved, though is this known? Therefore, half-lives of mRNAs does not muddle the analysis.

9) In the Discussion section, you state that “DSB resection not only generates ssDNA ends necessary to initiate HR, but can also facilitate the access of DNA repair proteins to the site of damage by releasing the transcriptional machinery from the DSB. That is speculation, right? Label as such.

*Reviewer #3*:

In the manuscript by Manfrini et al, the authors examine the transcript levels of genes flanking a DNA break in budding yeast and find that transcript levels decrease over time. The decrease in transcription correlates with resection levels, and strains with altered kinetics of resection show corresponding altered kinetics of transcription changes.

The data in the manuscript are generally of good quality and the changes in transcription and resection are very clear. The really major problem with this manuscript is that the conclusions are written in such a way as to imply that resection is regulating transcriptional repression. However, the data in the manuscript very clearly demonstrates that at the 240 minute time point (at which most of the transcription assays are performed), there is extensive resection of the area containing the genes being analyzed. Resection of double-stranded DNA will prevent transcription factors (such as TBP) from binding to promoter elements and as a result, transcription is no longer possible (and of course, depending on the orientation of the gene, resection can mean that there is no longer a template for the RNA polymerase!). Consequently, most of the cells in the population are unable to carry out transcription of these genes at this time point. This is quite different from ‘transcriptional repression’, which implies regulation. At earlier time points (e.g. shown in Figure 1), the only genes affected are precisely those genes that would no longer be completely double stranded in many cells. The ChIP data in Figure 4 fully support this (i.e. that loss of the double stranded DNA sequence results in loss of binding by RNA polymerase II).

Therefore, what the manuscript is in fact demonstrating is that transcription is never repressed in yeast in genes flanking a DNA break. It is only impeded by loss of DNA sequence during resection. This is different from what has been established in the mammalian system, where there is an active mechanism for signalling to the transcriptional machinery over multiple kilobases where resection isn't taking place. Therefore, if the manuscript is rewritten to make this clear it would be of interest.

---

## [Author Response]

*1) We all felt that the likely reason why transcription stops and RNA polymerase II occupancy is lost upon resection is because the transcription machinery, such as TBP and RNA polymerase II bind to dsDNA and not ssDNA. This stopping transcription due to loss of DNA template model is quite a different model from the current repression model, which implies an active regulatory mechanism. As such, an in depth discussion of loss of transcription as a consequence of loss of DNA template with appropriate citations, and a model figure to illustrate this idea, is needed*.

We have now provided a discussion of loss of transcription as consequence of loss of DNA template with appropriate references. Furthermore, we now show a model in new Figure 7 to illustrate this idea.

*2) The data generated from the HO site at* LEU2 *and the HO site at* MAT *need to be in two separate figures, and the difference emphasized more strongly in the text. Otherwise this will all be assumed to be at* MAT *and the apparent novelty of this work, compared to the previous Lee et al., paper, will be reduced*.

We now show the data regarding the HO site at the *MAT* and *LEU2* loci in two separate figures (Figures 2 and 3). To emphasize that DSB-induced transcriptional inhibition is not locus specific, we have analyzed transcription around a third DSB at the *ARG5,6* locus (see point 1, reviewer 2). The data regarding loss of transcription around the DSB at the *LEU2* and *ARG5,6* loci are reported in the new Figure 3.

*3) The result with the* mec1∆ tel1∆ *double mutant needs to be reported in order to conclude that the DNA damage checkpoint is not required for stopping transcription around a DNA double strand break*.

We have repeated the analysis in *mec1∆ tel1∆* double mutant. The double mutant represses transcription around the DSB like each single *mec1∆* and *tel1∆* mutant. These results are reported in new Figure 4.

Reviewer #1:

*Although brief, this is an important study that makes significant advances on the previous work from Jim Haber's lab showing that there is progressive transcriptional repression around the DNA double strand break (DSB) induced by the HO endonuclease at the yeast* MAT *locus. There are two important novel results from the current work: 1) the transcriptional repression around a DSB is not dependent on checkpoint activation (in contrast to the situation in mammalian cells); 2) transcriptional repression is dependent on DNA resection. This last finding is very novel and exciting, and has not yet been examined in other systems. The work is of high quality, but the following point need to be addressed*:

*Given the partial redundancy of Tel1 and Mec1 in activation of the DNA damage checkpoint, it is important that they show that transcriptional repression around the HO site is not affected in a* tel1 mec1 sml1 *mutant, in order to make the conclusion that the activation of the DNA damage checkpoint is not required for transcriptional repression*.

We have repeated the analysis in *mec1∆ tel1∆* double mutant. The double mutant represses transcription around the DSB like each single *mec1∆* and *tel1∆* mutant. These results are reported in new Figure 4.

*Mechanically, it makes sense that DNA resection would block transcription, given that RNA polymerase II binds to double strand DNA, and cannot transcribe from single strand DNA. This should be discussed briefly*.

We have discussed this point.

Reviewer #2:

*1) They need to emphasize that they carried out this analysis on a non-*MAT *locus DSB,*
Figure 2
*is of an HO break at the* LEU2 *locus. So, they have tested 2 DSBs. Is their general conclusion made any more convincing by test of a third DSB? Yet, showing data at the HO site at* LEU2 *is critical for the generality of their conclusions. I suggest they therefore put the* LEU2 *HO data in a separate Figure. Shape the data in a similar fashion as the HO data, state distance from HO site, extending left and right from the DSB, as they did for the* MAT *HO site data. Parallel presentations make it easier for the reader*.

We now show the data regarding the HO site at the *MAT* and *LEU2* loci in two separate figures (new Figures 2 and 3). To emphasize that DSB-induced transcriptional inhibition is not locus specific, we have analyzed transcription around a third DSB at the *ARG5,6* locus. The data regarding loss of transcription around the DSB at the *LEU2* and *ARG5,6* loci are reported in new Figure 3.

*2) As I said in the general comments, they need to show a model. The ‘take home’ message is simple, and easily grasped by all with a model. That model can have Rad9 and H2A∆ in it. Yet, most critically is the double-strand to single-strand conversion as the mechanistic explanation. And this inhibition appears to be perhaps unintended consequences of a separate biological reaction, generating a substrate for recombination*.

We now show a model in new Figure 7.

*3) They need a citation that in fact single-strand templates do not support transcription*.

We have now provided a discussion of loss of transcription as consequence of loss of DNA template with appropriate references. Furthermore, we now show a model in new Figure 7 to illustrate this idea.

*4) I do not quite get how the transcriptional repression cannot be Mec1 regulated, if Mec1 regulates degradation by regulating Rad9, etc.? And degradation of telomere proximal DNA has complex interactions with Mec1 and Rad53, and Rad9; a word on how the telomere resection is or is not relevant here*.

We have previously shown that cells lacking Mec1 accelerates DSB resection because they fail to recruit Rad9 to the DSB (5). However, DSB resection in *mec1* mutant cells is not as fast as in *rad9* cells because Mec1 has also a positive role in promoting resection by phosphorylating and activating Sae2 (5). In any case, consistent with a slight acceleration of DSB resection, the decrease in mRNA levels in *mec1* mutant cells in Figure 4 is slightly more pronounced than in wild type.

We prefer not discussing resection at telomeres, because it is quite limited compared to that at DNA DSBs and we do not know the impact on transcription.

*5) In the Results section, the authors allude to a previous paper (*[24]*, EMBO Rep) stating: “We previously used this strain for total RNA-seq analysis of protein-coding gene expression upon induction of the HO-induced DSB”*. *What did they conclude from that study relevant to this study?*

We have modified the sentence*.*

*6) The conclusion that the inhibition is not cell cycle regulated, based on but two experiments – asynchronous and G2/M synchronized – is underwhelming. At least under those two conditions they see no difference; tone down the generalization, as the experiments do not support that conclusion. I do not suggest getting rigorous about the cell cycle effects, or lack thereof; this is not an important issue, it seems. Just have the language match the data in this point*.

We have deleted the paragraph.

*7) Make it more transparent how gene transcription reduction correlates with extent of degradation. So, when reporting on the r2 restriction site, state how far away it is from the HO site, what gene is there, and how much that genes transcription is reduced and at what time*.

We have modified the text accordingly.

*8) At the Results outset, I think it best to set the reader up for the story and technical issues. It is relevant that the time scale, in hours, is far longer than the half-lives of the mRNAs involved, though is this known? Therefore, half-lives of mRNAs does not muddle the analysis*.

We have added a sentence that the RNA decreases at 240 minutes after HO induction were not influenced by mRNA stability, as all the analyzed mRNAs have half-lives shorter than 50 minutes (10).

*9) In the Discussion section, you state that “DSB resection not only generates ssDNA ends necessary to initiate HR, but can also facilitate the access of DNA repair proteins to the site of damage by releasing the transcriptional machinery from the DSB. That is speculation, right? Label as such*.

We have modified the sentence.

Reviewer #3:

*[…] The data in the manuscript are generally of good quality and the changes in transcription and resection are very clear. The really major problem with this manuscript is that the conclusions are written in such a way as to imply that resection is regulating transcriptional repression. However, the data in the manuscript very clearly demonstrates that at the 240 minute time point (at which most of the transcription assays are performed), there is extensive resection of the area containing the genes being analyzed. Resection of double-stranded DNA will prevent transcription factors (such as TBP) from binding to promoter elements and as a result, transcription is no longer possible (and of course, depending on the orientation of the gene, resection can mean that there is no longer a template for the RNA polymerase!). Consequently, most of the cells in the population are unable to carry out transcription of these genes at this time point. This is quite different from ‘transcriptional repression’, which implies regulation. At earlier time points (e.g. shown in*
Figure 1*), the only genes affected are precisely those genes that would no longer be completely double stranded in many cells. The ChIP data in*
Figure 4
*fully support this (i.e. that loss of the double stranded DNA sequence results in loss of binding by RNA polymerase II)*.

*Therefore, what the manuscript is in fact demonstrating is that transcription is never repressed in yeast in genes flanking a DNA break. It is only impeded by loss of DNA sequence during resection. This is different from what has been established in the mammalian system, where there is an active mechanism for signalling to the transcriptional machinery over multiple kilobases where resection isn't taking place. Therefore, if the manuscript is rewritten to make this clear it would be of interest*.

We have modified the title and the text to make the point raised by the reviewer clear.